# Accelerated fetal growth velocity across the third trimester is associated with increased shoulder dystocia risk among fetuses who are not large-for-gestational-age: A prospective observational cohort study

**Teresa M. MacDonald**[1,2]*, **Alice J. Robinson**[1], **Richard J. Hiscock**[2], **Lisa Hui**[1,2], **Kirsten M. Dane**[1], **Anna L. Middleton**[1,2], **Lucy M. Kennedy**[2], **Stephen Tong**[1,2], **Susan P. Walker**[1,2]

**1** Mercy Perinatal, Mercy Hospital for Women, Melbourne, Victoria, Australia, **2** Department of Obstetrics and Gynaecology, University of Melbourne, Melbourne, Victoria, Australia

* teresa.mary.macdonald@gmail.com

## Abstract

### Objective

To investigate whether fetuses with accelerated third trimester growth velocity are at increased risk of shoulder dystocia, even when they are not large-for-gestational-age (LGA; estimated fetal weight (EFW) >95th centile).

### Methods

Fetal growth velocity and birth outcome data were prospectively collected from 347 nulliparous women. Each had blinded ultrasound biometry performed at 28 and 36 weeks' gestation. Change in EFW and abdominal circumference (AC) centiles between 28–36 weeks were calculated, standardised over exactly eight weeks. We examined the odds of shoulder dystocia with increasing EFW and AC growth velocities among women with 36-week EFW ≤95th centile (non-LGA), who went on to have a vaginal birth. We then examined the relative risk (RR) of shoulder dystocia in cases of accelerated EFW and AC growth velocities (>30 centiles gained). Finally, we compared the predictive performances of accelerated fetal growth velocities to 36-week EFW >95th centile for shoulder dystocia among the cohort planned for vaginal birth.

### Results

Of the 226 participants who had EFW ≤95th centile at 36-week ultrasound and birthed vaginally, six (2.7%) had shoulder dystocia. For each one centile increase in EFW between 28–36 weeks, the odds of shoulder dystocia increased by 8% (odds ratio (OR [95% Confidence Interval (CI)]) = 1.08 [1.04–1.12], p<0.001). For each one centile increase in AC between 28–36 weeks, the odds of shoulder dystocia increased by 9% (OR[95%CI] = 1.09 [1.05–1.12], p<0.001). When compared to the rest of the cohort with normal growth velocity,

**Data Availability Statement:** All relevant data are within the manuscript and its Supporting Information files.

**Funding:** National Health and Medical Research Council (NHMRC; www.nhmrc.gov.au) Grant (#1065854) to SW. and an Australian Government Research Training Program Scholarship (www.education.gov.au/research-training-program) to TM. Funding sources had no involvement in study design, collection or analysis of data, decision to publish, or in the writing or submission of this manuscript.

**Competing interests:** The authors have declared that no competing interests exist.

accelerated EFW and AC velocities were associated with increased relative risks of shoulder dystocia (RR[95%CI] = 7.3 [1.9–20.6], $p$ = 0.03 and 4.8 [1.7–9.4], $p$ = 0.02 respectively). Accelerated EFW or AC velocities predicted shoulder dystocia with higher sensitivity and positive predictive value than 36-week EFW >95th centile.

## Conclusions

Accelerated fetal growth velocities between 28–36 weeks' gestation are associated with increased risk of shoulder dystocia, and may predict shoulder dystocia risk better than the commonly used threshold of 36-week EFW >95th centile.

## Introduction

Macrosomia (birthweight >4000g) is the most significant risk factor for shoulder dystocia [1], which occurs more frequently with increasing birthweight [1–5]. Shoulder dystocia complicates between 0.2% and 3.0% of all vaginal deliveries [6] and is associated with brachial plexus injury, fractures, and birth asphyxia for the infant; and increased maternal haemorrhage and genital tract trauma [5,7]. However, clinicians have an opportunity to intervene and reduce these risks. A recent systematic review has reported that induction of labour in cases of suspected macrosomia (estimated fetal weight (EFW) >95th centile [8]) reduces both shoulder dystocia and fracture risk, compared to expectant management [9].

Problematically, only half of all shoulder dystocias occur in infants with birthweight >4000g [1,5]. Thus, shoulder dystocia remains a largely unpredictable event, with known risk factors consistently demonstrating poor positive predictive value [1,5]. Improved predictive tools would enable clinicians to counsel and manage pregnant women appropriately, and reduce the risk of this significant obstetric emergency. Prediction and prevention of shoulder dystocia, particularly among fetuses that are not large-for-gestational-age (LGA; EFW >95th centile), remains an unmet gap in clinical care.

Recently, we found that fetuses with reduced growth velocity (a fall in EFW or abdominal circumference (AC) centile across the third trimester) demonstrate evidence of placental insufficiency even when they are not small-for-gestational-age (<10th centile) at birth [10]. Specifically, we discovered that the lower the fetal growth velocity, the lower the neonatal fat stores [10]. This means that the corollary also holds true: the higher the fetal growth velocity, the higher the neonatal fat stores [10].

We hypothesised that if reduced fetal growth velocity represents relative placental insufficiency (growth less than expected by genetic potential) among fetuses that are not small [10], accelerated fetal growth velocity might indicate pathological overgrowth (growth in excess of the genetic growth potential) even where the EFW is ≤95th centile. If so, we might expect those with accelerated fetal growth velocity to exhibit increased shoulder dystocia risk. Here, we investigate whether accelerated fetal growth velocity between 28 and 36 weeks is associated with shoulder dystocia among fetuses who are not LGA.

## Materials and methods

### Study design overview

This study analysed data from participants of the Fetal Longitudinal Assessment of Growth (FLAG) study. The FLAG study was a prospective longitudinal study conducted at the Mercy

Hospital for Women, a tertiary maternity hospital in Melbourne with approximately 6000 births each year.

As previously described [10], ultrasound biometry was prospectively used to measure EFW and AC at 28 and 36 weeks' gestation. For each of these ultrasound examinations, the gestation-dependent EFW and AC centiles were determined. The associations between relative EFW and AC centile change between 28–36 weeks and the occurrence of shoulder dystocia were evaluated.

This study was designed to investigate whether fetuses that showed an increase in growth velocity across the third trimester had an increased risk of shoulder dystocia, even when not LGA. LGA fetuses (EFW >95th centile [8]) are already known to be at increased risk, and to benefit from induction of labour [9]. As such, cases where the EFW was >95th centile at the time of the 36-week ultrasound scan were excluded from the primary analysis. Similarly, since shoulder dystocia is a complication of vaginal birth, infants delivered by caesarean section were excluded.

This study was approved by Mercy Health Research Ethics Committee, Ethics Approval Number R14/12, and written informed consent was obtained from all participants.

## Determination of EFW and AC centiles

28- and 36-week research ultrasound examinations were performed by one of two experienced operators (TMM and AJR) between 27+0 and 29+0 weeks, and 35+0 and 37+0 weeks' gestation respectively, as previously described [10]. Following delivery, ultrasound EFWs and birthweights were customised using the GROW software [11] (http://www.gestation.net/). We adjusted for maternal height, weight and parity, fetal/infant sex, and exact gestational age. We did not customise for ethnicity. AC measurements were converted to a z-score, then centile, using the Chitty AC equation [12].

Importantly, treating clinicians were blinded to 28- and 36-week ultrasound results. They were not informed of the EFW or AC centile, unless the EFW was below the 10th centile. EFWs >95th centile, and EFW and AC changes in centile between ultrasound examinations, were not disclosed. There was good inter- and intra-observer agreement for ultrasound biometric measures, as previously reported [10].

## Calculating fetal growth velocity

To determine the EFW growth velocities, we calculated the change in EFW centile between 28–36 weeks by subtracting the 28-week customised EFW centile from the 36-week customised EFW centile. The same process was undertaken to calculate AC growth velocity. A fetus whose EFW or AC increased over time thus had a positive number to describe the EFW or the AC growth velocity. A fetus with no change in centile between ultrasounds had a growth velocity of zero, and a fetus whose EFW or AC centile reduced between ultrasounds had negative growth velocity values. To ensure that comparison of growth velocity was standardised for the cohort, the change in EFW centile, and in AC centile, between the two ultrasounds were each divided by the exact number of days between examinations. This created a centile change per day, which was then multiplied by 56 to facilitate comparison of individualised centile change over exactly eight weeks.

## Birthing outcome data and definition of shoulder dystocia

Delivery outcomes were reviewed by a single clinician, blinded to ultrasound growth velocity results. The diagnosis of shoulder dystocia was made where the midwife or doctor present at the birth had documented shoulder dystocia in the medical record as well as at least one

manoeuvre to manage it. The manoeuvres include one or more of McRoberts, Rubin I, Rubin II, woodscrew, reverse woodscrew, delivery of the posterior arm, or change of maternal position onto all fours.

## Statistical analysis

Consistent with our previous study [10] we primarily defined accelerated growth velocity as an increase in EFW of >30 centiles between the two ultrasound scans. Maternal characteristics and birth outcome data were compared between cases of accelerated (>30 centiles over exactly eight weeks), versus normal (≤30 centiles) EFW growth velocity. To do this we used unpaired t-test (if normally distributed) or Mann-Whitney test (if not normally distributed) for continuous data, and chi-squared or Fisher's exact tests for categorical analyses.

We assessed the relationships between EFW and AC growth velocities and shoulder dystocia in two ways: (i) we analysed the growth velocities as continuous variables against the outcome of interest–shoulder dystocia–using logistic regression. This determined the odds of shoulder dystocia per centile increase in 28–36 week EFW or AC growth velocity; and (ii) we assessed our pre-defined dichotomous thresholds, increase of EFW and AC growth velocities of >30 centiles over eight weeks compared to the remainder of the cohort, using Fisher's exact test. This ascertained the relative risk (RR) of shoulder dystocia where an increase in centile of this magnitude was seen. Finally we compared the predictive performances of accelerated 28–36 week EFW and AC growth velocities (>30 centiles) and 36-week EFW >95th centile for shoulder dystocia. These performance parameters were calculated from all participants planned for vaginal delivery–replicating the cohort at potential risk of shoulder dystocia when seen in the antenatal clinic.

Statistical analysis was performed using GraphPad Prism versions 6.00 and 8.00 for Windows (GraphPad Software, La Jolla, CA, USA, http://www.graphpad.com/), except for logistic regression, which was performed using Stata Statistical Software Release 16 (College Station, TX, USA).

## Results

### Study participants

Between February 2015 and February 2016, 347 participants completed both study ultrasounds allowing calculation of third trimester fetal growth velocities. There were 39 (11.2%) participants with an estimated fetal weight (EFW) >95th centile at the time of the 36-week ultrasound scan. 36-week EFW >95th centile performed well in this cohort for the prediction of birthweight >95th centile. It demonstrated sensitivity of 81.8% [95% confidence interval = 48.2%-97.7%], 91.1% [87.5%-93.9%] specificity, a positive predictive value of 23.1% [16.2%-31.8%], and 99.4% [98.8%-99.8%] negative predictive value. Of the 308 participants not suspected to carry an LGA fetus, 82 (26.6%) delivered by caesarean section. Of the 226 (73.4%) participants with EFW ≤95th centile who birthed vaginally, there were six (2.7%) cases of shoulder dystocia (Fig 1). We compared their baseline maternal characteristics and pregnancy outcomes to those EFW ≤95th centile cases who birthed vaginally without shoulder dystocia occurring (Table 1).

While birthweight and birthweight centile were slightly higher among those who had shoulder dystocia, these did not reach statistical significance. Women whose births were complicated by shoulder dystocia were a mean 4.5 years older, and had median higher booking BMI than those who birthed without shoulder dystocia. The 28–36 EFW and AC growth velocities were significantly higher, as were the proportions of shoulder dystocia cases who had demonstrated accelerated (>30 centiles gained) EFW or AC velocity between 28 and 36 weeks.

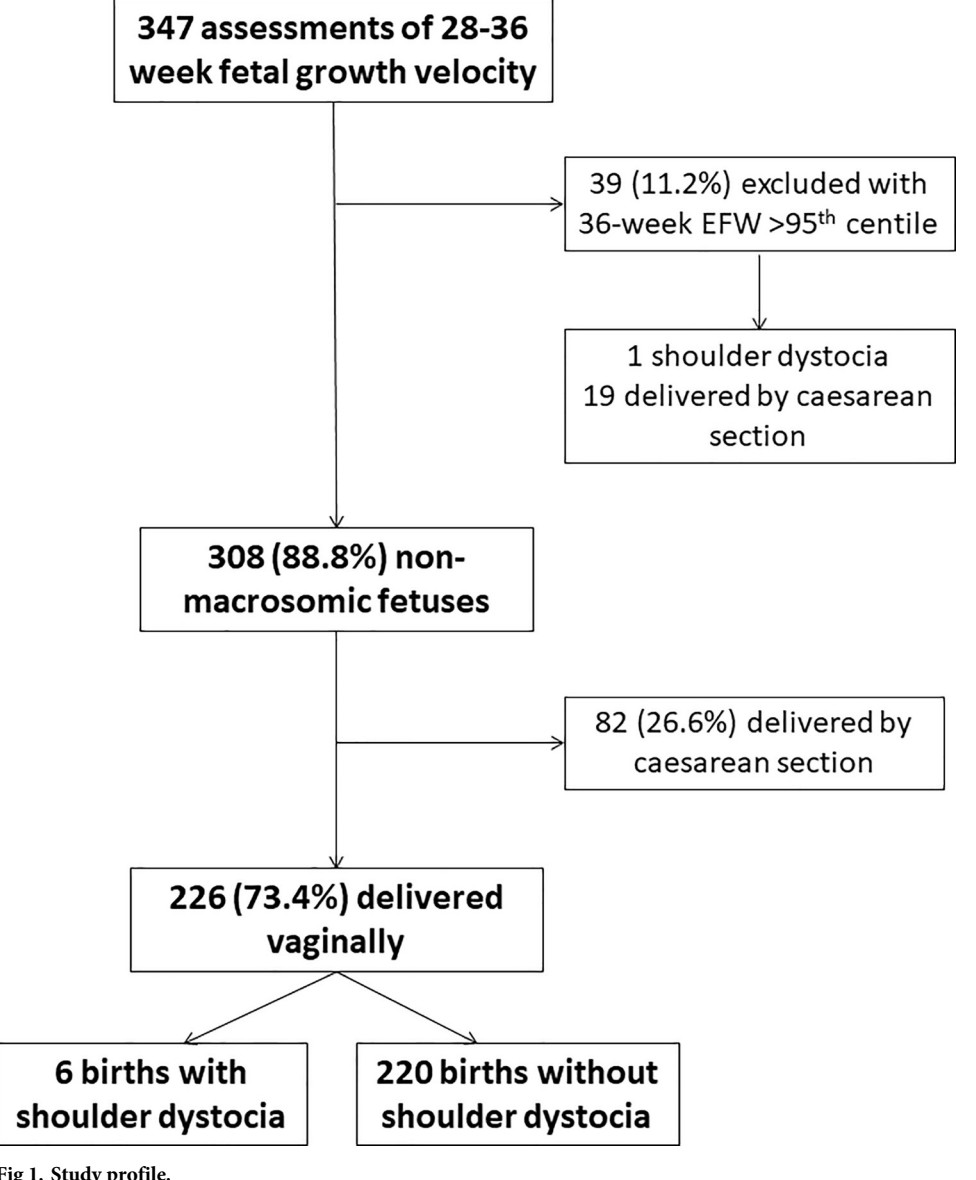

**Fig 1. Study profile.**

### 28–36 week EFW and AC growth velocities and shoulder dystocia

We then performed multivariate logistic regression to examine the relationship between 28–36 week EFW and AC growth velocities among non-LGA fetuses and shoulder dystocia, accounting for both maternal age and booking BMI as potential confounders. For every one centile increase in EFW over eight weeks, the odds of shoulder dystocia increased by 8% (Odds ratio (OR) [95% confidence interval (CI)] = 1.08 [1.04–1.12], $p < 0.001$). For every one centile increase in AC over eight weeks, the odds of shoulder dystocia increased by 9% (OR [95% CI] = 1.09 [1.05–1.12], $p < 0.001$).

We then examined the risk of shoulder dystocia if the cohort was dichotomised according to clinical thresholds of 28–36 week EFW and AC growth velocities of >30 centiles over eight weeks. 12 fetuses demonstrated accelerated EFW growth velocity, and 26 demonstrated

**Table 1. Maternal characteristics and delivery outcomes for cases of shoulder dystocia compared to the rest of the cohort.**

|  | Shoulder dystocia (n = 6) | No shoulder dystocia (n = 220) | *p* |
|---|---|---|---|
| **Age (years)** | 34.8 (4.5) | 30.3 (3.7) | 0.003 |
| **Booking BMI (kg/m$^2$)** | 27.4 [25.1–28.8] | 23.3 [21.2–26.8] | 0.02 |
| **Current smoking** | 1 (16.7%) | 3 (1.4%) | 0.10 |
| **Preeclampsia** | 1 (16.7%) | 11 (5.0%) | 0.28 |
| **Gestational diabetes** | 1 (16.7%) | 20 (9.1%) | 0.45 |
| **28–36 week EFW velocity** | 22.6 (23.2) | -8.0 (21.0) | 0.0005 |
| **28–36 week AC velocity** | 29.6 (16.8) | -0.7 (23.3) | 0.002 |
| **Accelerated 28–36 week EFW velocity** | 2 (33.3%) | 10 (4.5%) | 0.03 |
| **Accelerated 28–36 week AC velocity** | 3 (50.0%) | 23 (10.5%) | 0.02 |
| **Induction of labour** | 4 (66.7%) | 109 (49.5%) | 0.68 |
| **Instrumental delivery** | 5 (83.3%) | 100 (45.5%) | 0.10 |
| **Birthweight (g)** | 3507 (460) | 3292 (421) | 0.22 |
| **Birthweight centile** | 54.3 [35.1–61.4] | 34.6 [16.3–58.3] | 0.14 |
| **Birthweight >4000g** | 1 (16.7%) | 14 (6.4%) | 0.34 |
| **Gestation at birth (weeks)** | 40.0 [39.0–41.0] | 40.0 [39.0–40.6] | 0.75 |

The cohort summarised in Table 1 includes all women who delivered their infants vaginally, and where the EFW was ≤95th centile at 36-week ultrasound. Data presented as mean (standard deviation) if normally distributed, as median [interquartile range] if not normally distributed, and as number (%) if categorical. AC = Abdominal circumference; BMI = Body Mass Index; EFW = Estimated fetal weight. "Accelerated 28–36 week velocities" refers to fetuses that demonstrated an increase in EFW or AC of more than 30 centiles between their 28 and 36 week ultrasound scans, standardised over exactly 8 weeks.

accelerated AC growth velocity. Shoulder dystocia occurred in 2/12 (17%) of births in those with accelerated EFW growth velocity compared to 4/214 (2%) among those with normal growth velocity (RR [95% CI] 7.3 [1.9–20.6], *p* = 0.03). Similarly, there was almost five times greater risk of shoulder dystocia for fetuses with accelerated 28–36 week AC growth velocity (Table 2).

## Accelerated fetal growth velocities compared to EFW >95th centile as predictors of shoulder dystocia

36-week ultrasound EFW >95th centile is currently clinically used as a threshold at which to recommend induction of labour between 37–38+6 weeks in order to reduce shoulder dystocia [8]. Given this, we compared the performances of accelerated 28–36 week EFW and AC growth velocities with 36-week EFW >95th centile in predicting shoulder dystocia. We included the whole cohort planned for vaginal birth–regardless of eventual mode of delivery– when calculating the predictive performance parameters for accelerated EFW growth velocity, accelerated AC growth velocity, and 36-week EFW >95th centile. This was to replicate

**Table 2. Relative risk of shoulder dystocia in cases of accelerated (>30 centiles gained) 28–36 week fetal growth velocity.**

| Growth parameter | Shoulder dystocia n(%) | | RR (95% CI) if high velocity | *p* |
|---|---|---|---|---|
|  | Accelerated growth velocity | Normal growth velocity |  |  |
| EFW | 2/12 (16.7%) | 4/214 (1.9%) | 7.3 (1.9–20.6) | 0.03 |
| AC | 3/26 (11.5%) | 3/200 (1.5%) | 4.8 (1.7–9.4) | 0.02 |

Relative risks (RR) calculated among infants born vaginally, with EFW ≤95th centile at 36-week ultrasound scan. AC = abdominal circumference; EFW = estimated fetal weight; CI = confidence interval.

**Table 3. Maternal characteristics and delivery outcomes for cohort of participants planned for vaginal delivery, and comparison of EFW >95th centile group to EFW ≤95th centile group at 36-week ultrasound.**

| | Total cohort (n = 318) | 36 week ultrasound EFW >95th centile (n = 36) | 36 week ultrasound EFW ≤95th centile (n = 282) | P |
|---|---|---|---|---|
| Age (years) | 30·7 (3·9) | 31.9 (3.9) | 30.5 (3.9) | 0.05 |
| Booking body mass index (kg/m²) | 23·6 [21·4–26·7] | 23.2 [21.3–25.6] | 23.7 [21.4–26.9] | 0.39 |
| Smokers | 6 (1.9%) | 1 (2.8%) | 5 (1.8%) | 0.52 |
| Preeclampsia | 19 (5.9%) | 2 (5.6%) | 17 (6.0%) | 1.00 |
| Gestational diabetes | 37 (11.6%) | 3 (8.3%) | 34 (12.1%) | 0.78 |
| Induction of labour | 170 (53.5%) | 18 (50.0%) | 152 (53.9%) | 0.72 |
| Mode of birth | | | | |
| Normal vaginal birth | 131 (41.2%) | 10 (27.8%) | 121 (42.9%) | 0.004 |
| Instrumental birth | 115 (36.2%) | 10 (27.8%) | 105 (37.2%) | |
| Unscheduled caesarean | 72 (22.6%) | 16 (44.4%) | 56 (19.9%) | |
| Shoulder dystocia | 7 (2.2%) | 1 (2.8%) | 6 (2.1%) | 0.57 |
| Birthweight (g) | 3375 (471) | 3872 (334) | 3311 (448) | <0.0001 |
| Birthweight centile | 39.3 [18.7–67.1] | 85.5 [74.2–94.2] | 35.3 [17.4–59.5] | <0.0001 |
| Birthweight >4000g | 33 (10.4%) | 13 (36.1%) | 20 (7.1%) | <0.0001 |
| Gestational age at birth (weeks) | 40.0 [39.0–40.7] | 39.8 [38.9–40.6] | 40.0 [39.0–40.7] | 0.88 |

Data presented as mean (standard deviation) if normally distributed, as median [interquartile range] if not normally distributed, and as number (%) if categorical. EFW = Estimated fetal weight.

application of this information in the antenatal clinic where mode of delivery is not yet known. 29 (8.4%) women planned for pre-labour elective caesarean section were thus excluded.

First, we compared the maternal characteristics and birth outcomes of those with an LGA fetus at scan to the rest of the cohort planned for vaginal birth. Participants with 36-week EFW >95th centile were significantly more likely to be delivered by unscheduled caesarean section and to deliver larger neonates (Table 3). Summarised in Table 4, accelerated fetal growth velocities were stronger predictors of shoulder dystocia than 36-week ultrasound EFW >95th centile, with better performance than classification as LGA in every category. Accelerated EFW growth velocity demonstrated the highest positive predictive value (12.5%), specificity (95.5%) and positive likelihood ratio (6.4), while accelerated AC growth velocity demonstrated the

**Table 4. Performance of accelerated 28–36 week fetal growth velocities (>30 EFW/AC centiles) and 36-week ultrasound EFW >95th centile in predicting shoulder dystocia among women planned for vaginal delivery.**

| | Accelerated EFW growth velocity | Accelerated AC growth velocity | 36-week ultrasound EFW >95th centile |
|---|---|---|---|
| Number (%) | 16 (5.0%) | 42 (13.2%) | 36 (11.3%) |
| Sensitivity | 28.6% (3.7% - 71.0%) | 42.9% (9.9% - 81.6%) | 14.3% (0.4% - 57.9%) |
| Specificity | 95.5% (92.6% - 97.5%) | 87.5% (83.3% - 90.9%) | 88.8% (84.7% - 92.0%) |
| Positive likelihood ratio | 6.4 (1.8–22.8) | 3.4 (1.4–8.4) | 1.3 (0.2–8.0) |
| Negative likelihood ratio | 0.8 (0.5–1.2) | 0.7 (0.3–1.2) | 1.0 (0.7–1.3) |
| Positive predictive value | 12.5% (3.8% - 33.9%) | 7.1% (3.0% - 16.0%) | 2.8% (0.5% - 15.3%) |
| Negative predictive value | 98.3% (97.4% - 99.0%) | 98.6% (97.3% - 99.2%) | 97.9% (97.1% - 98.4%) |

There were 318 women planned for vaginal birth in total. Ranges within brackets represent 95% confidence intervals. EFW = estimated fetal weight, AC = abdominal circumference.

highest sensitivity (42.9%), and negative predictive value (98.6%). 36-week EFW >95[th] centile performed with a particularly low positive predictive value, of only 2.8%.

## Discussion

We report for the first time that fetuses with accelerated third trimester fetal growth velocity are at increased risk of shoulder dystocia, even when they are not LGA. The odds of shoulder dystocia rise with increasing EFW and AC centile between 28–36 weeks. Clinically relevant thresholds of accelerated fetal growth velocity, >30 EFW or AC centiles gained over eight weeks, correspond to clinically significant increased absolute, and relative, risks for shoulder dystocia. Further, in our cohort, accelerated growth velocities demonstrate better predictive performance for shoulder dystocia than EFW >95[th] centile–a threshold currently used to recommend earlier term induction [8,9]. Accelerated fetal growth velocity could thus potentially alert clinicians of otherwise unsuspected increased shoulder dystocia risk, improving the prediction and prevention of, this serious obstetric emergency.

New and better tools to predict shoulder dystocia risk are a pressing clinical need. Even a landmark legal case [13] has highlighted that, given the potential severe consequences of shoulder dystocia, the onus is on clinicians to provide pregnant women with risk assessment and options regarding mode of birth. Evidence is accumulating for the safety and efficacy of early term induction to reduce shoulder dystocia risk [9] which may be preferred by clinicians and women to elective caesarean given ultrasound has poorer accuracy, and higher false positive rates, in large babies [14–16]. Induction is not associated with a significant increase in emergency caesarean section [9]. Induction of labour for LGA fetuses (clinical suspicion followed by EFW >95[th] centile), represents the only risk factor and intervention dyad shown in a randomised controlled trial to reduce shoulder dystocia [8]. That accelerated fetal growth velocities had superior predictive performances than 36-week EFW >95[th] centile in our cohort raises the tantalising possibility that a better parameter to identify shoulder dystocia risk may be available. That approximately half of shoulder dystocia cases occur in infants of <4000g birthweight [1,5] highlights the need for a useful predictor for non-LGA fetuses. The ability to offer timely induction of labour to women at increased risk of shoulder dystocia, even those carrying a non-macrosomic baby, could significantly improve obstetric care.

Unfortunately, accurate tools to predict shoulder dystocia and related birth injury are currently lacking. Previous shoulder dystocia engenders a heightened clinical alert, but risk factors are poor predictors of recurrence even among these at-risk women [17]. The most important risk factor is increased infant birthweight compared to the index pregnancy–even in the absence of macrosomia [17]. That this risk factor is obviously only applicable to parous women is a significant limitation.

Clinical, then ultrasound, evaluation of fetal size is the most commonly used triage to flag shoulder dystocia risk. While assessment for macrosomia is applicable to nulliparous and parous women alike, this approach has several limitations. There is no universally accepted ultrasound definition of macrosomia; there are multiple different formulae available from which to calculate EFW and by which to assign EFW centile; ultrasound biometry measurements all have clinically significant error margins; and ultrasound quality and formulae are both less accurate when imaging larger babies [16,18,19]. A systematic review of the many different ultrasound definitions of macrosomia recently evaluated absolute measurement and centile cut-offs for AC and EFW as well as other fetal measurements for their prediction of shoulder dystocia. The best ultrasound predictor of shoulder dystocia was not EFW >95[th] centile or >4000g, but was difference in abdominal and biparietal diameters of ≥2.6cm [18]. Definitions of macrosomia significantly associated with shoulder dystocia still failed to predict up to 60%

of cases [18]–again highlighting that a large proportion of dystocias occur among non-macrosomic infants. We suggest that accelerated growth velocity–identifying pathological overgrowth–may be a useful adjunct to dichotomous thresholds of suspected macrosomia in identifying pregnancies at risk. That the best current predictor of shoulder dystocia is a disproportionately large AC [18]–and that the AC is where accelerated fetal growth is most likely to manifest [20]–suggests that growth velocity, alone or in combination with other markers, has potential to improve predictive accuracy.

Two previous studies have evaluated accelerated fetal growth velocity, but did not find a significant association with shoulder dystocia [21,22]. This is likely due to differences in methodology compared to our study. First, both studied high-risk cohorts–women with impaired glucose tolerance–already known to be at increased risk of shoulder dystocia [5]. This raises the possibility of intervention bias through clinical care. This is evidenced in one study, where the participants were delivered much earlier than the women in our study–at a mean gestation of 37 weeks; and a very high proportion (27%) delivered infants with birthweights >95[th] centile [23]. Secondly, fetal growth velocities were likely calculated over shorter periods than eight weeks. One calculated growth velocities over three-five weeks [22,23], and the other calculated change in EFW per week from the final two growth scans (where the time interval was not reported) prior to delivery [21]. Further, the highest EFW growth velocity reported was >2% per week increase in EFW centile–equating to only 16% over eight weeks [21]. We suspect that a three-five week gestational epoch may be too short–and >2% per week too low an increase in centile–to demonstrate meaningful changes in growth velocity.

Strengths of this study include that it was prospective and blinded, minimising the risk of intervention bias, and that we applied a robust and clinically significant definition of shoulder dystocia. We specifically did not include a time-based definition of >60 seconds between delivery of the infant's head and body as this potentially defines over 10% of vaginal births as complicated by shoulder dystocia [6,24]. Instead, our shoulder dystocia rates (2.0% of pregnancies and 2.8% of vaginal births) are in keeping with its reported incidence [6]. That our prevalence is at the higher end of the range may reflect the high quality of our obstetrics trainees and midwives, our regular simulation training which includes shoulder dystocia emergencies, and a culture of early recognition and recourse to emergency manoeuvres, as well as calling for help and clear documentation. We specifically analysed nulliparous women with non-LGA fetuses–the cohort most in need, given they have the least information about shoulder dystocia risk available to guide their clinicians. To our knowledge, this study is the first to evaluate growth velocity as a risk factor for shoulder dystocia specifically among fetuses that are not LGA.

With regard to study limitations: first, this is an ancillary post-hoc analysis of our prospective observational cohort which was primarily studied to investigate reduced fetal growth velocity and indicators of placental dysfunction [10]. Secondly, sample size is the main limitation of these findings. That our study was small, with only seven cases of shoulder dystocia, means it requires validation in a larger cohort in order to assess for important sequelae of shoulder dystocia such as brachial plexus injury, hypoxic ischaemic encephalopathy, or maternal anal sphincter injury or haemorrhage. Nevertheless, the findings are striking and biologically plausible, suggesting this may be a useful, novel predictor of this serious complication. Given the gravity of the potential consequences associated with shoulder dystocia, and the lack of predictive tools currently available to clinicians and patients, validating these findings should be considered an urgent priority. However, given our small number of cases, the need for validation in larger cohorts cannot be overemphasised and we would caution against clinical implementation at this stage where our findings are based on such a limited number of cases.

## Conclusions

Prediction and prevention of shoulder dystocia among fetuses who are not LGA represents a current gap in clinical care. We propose that assessment of third trimester fetal growth velocity may identify currently unsuspected infants at increased risk of shoulder dystocia. This finding warrants urgent further validation in a much larger cohort. If validated, our data may inform the design of an appropriately powered interventional study, to elucidate the value of timely and carefully supervised birth for fetuses with accelerated third trimester growth velocity in order to reduce the risk, and consequences, of this devastating obstetric emergency.

## Supporting information

**S1 File. Study dataset.**
(XLSX)

## Acknowledgments

We wish to thank Dr Elizabeth Lockie, the health information services, birth suite, postnatal ward, nursery, university research department, and perinatal medicine staff at the Mercy Hospital for Women for their assistance in conducting this study. We thank the philanthropic donors to the Mercy Health Foundation, and the Foundation itself, for their generous support.

## Author Contributions

**Conceptualization:** Teresa M. MacDonald, Stephen Tong, Susan P. Walker.

**Data curation:** Teresa M. MacDonald, Alice J. Robinson, Kirsten M. Dane, Anna L. Middleton, Lucy M. Kennedy.

**Formal analysis:** Teresa M. MacDonald, Richard J. Hiscock, Lucy M. Kennedy, Susan P. Walker.

**Funding acquisition:** Teresa M. MacDonald, Stephen Tong, Susan P. Walker.

**Investigation:** Teresa M. MacDonald, Lisa Hui, Stephen Tong, Susan P. Walker.

**Methodology:** Teresa M. MacDonald, Alice J. Robinson, Lucy M. Kennedy, Stephen Tong, Susan P. Walker.

**Project administration:** Teresa M. MacDonald, Alice J. Robinson, Kirsten M. Dane, Anna L. Middleton, Susan P. Walker.

**Resources:** Kirsten M. Dane, Stephen Tong, Susan P. Walker.

**Software:** Richard J. Hiscock.

**Supervision:** Alice J. Robinson, Lisa Hui, Stephen Tong, Susan P. Walker.

**Writing – original draft:** Teresa M. MacDonald.

**Writing – review & editing:** Alice J. Robinson, Richard J. Hiscock, Lisa Hui, Kirsten M. Dane, Anna L. Middleton, Lucy M. Kennedy, Stephen Tong, Susan P. Walker.

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
