## [Decision Letter · Decision Letter 0]

5 Aug 2021

PONE-D-21-00919

Accelerated fetal growth velocity across the third trimester is associated with increased shoulder dystocia risk among fetuses who are not large-for-gestational-age: A prospective observational cohort study

PLOS ONE

Dear Dr. MacDonald,

Thank you for submitting your manuscript to PLOS ONE. After careful consideration, we feel that it has merit but does not fully meet PLOS ONE’s publication criteria as it currently stands. Therefore, we invite you to submit a revised version of the manuscript that addresses the points raised during the review process.

Both reviewers recognized that this paper is original and interesting, thus, please carefully respond the critiques pointed out by them for further acceptance.

We look forward to receiving your revised manuscript.

Kind regards,

Kazumichi Fujioka

Academic Editor

PLOS ONE

Journal Requirements:

2. Please include captions for your Supporting Information files at the end of your manuscript, and update any in-text citations to match accordingly. Please see our Supporting Information guidelines for more information: http://journals.plos.org/plosone/s/supporting-information

Reviewers' comments:

Reviewer's Responses to Questions

**Comments to the Author**

1. Is the manuscript technically sound, and do the data support the conclusions?

Reviewer #1: No

Reviewer #2: Yes

2. Has the statistical analysis been performed appropriately and rigorously? 

Reviewer #1: No

Reviewer #2: Yes

3. Have the authors made all data underlying the findings in their manuscript fully available?

Reviewer #1: Yes

Reviewer #2: Yes

4. Is the manuscript presented in an intelligible fashion and written in standard English?

Reviewer #1: Yes

Reviewer #2: Yes

5. Review Comments to the Author

Reviewer #1: MacDonald et al present an interesting cohort study on the risk of shoulder dystocia according to accelerated fetal growth. I have the following comments/questions:

Abstract

Methods

Line 34: “We then compared the predictive 35 performances of accelerated EFW and AC growth velocities (>30 centiles gained) with 36-week 36 EFW >95th centile for shoulder dystocia among the cohort planned for vaginal birth”. This sentence is not clear. The authors’ analysis consisted in 2 steps, could they clarify them ?

Results

Why did the authors use both OR and RR to describe their results?

Introduction

The authors should mention the prevalence of shoulder dystocia

The following quote does not seem to be appropriate for an international audience: ine 66 “consistently demonstrating poor positive predictive value [1, 5]. That clinicians require better ways of identifying women at risk of shoulder dystocia is evidenced by the high profile case successfully lodged against the Lanarkshire Health Board after Nadine Montgomery’s son suffered cerebral palsy following shoulder dystocia”

Line 79 “Importantly, the corollary was also noted: the higher the fetal growth velocity, the higher the neonatal fat stores” Was this demonstrated in the authors’ previous publication

What do the authors mean by “pathological overgrowth” (line 84)? A fat distribution that mimics diabetes mellitus, increasing the risk of shoulder dystocia independently of the actual birthweight ?

Methods/Results

Line 165 “We assessed the relationships between EFW and AC growth velocities and shoulder dystocia in two ways: (i) we analysed the growth velocities as continuous variables against the outcome of interest – shoulder dystocia – using logistic regression. This determined the odds of shoulder dystocia per centile increase in 28-36 week EFW or AC growth velocity; and (ii) we assessed our pre-defined dichotomous thresholds, increase of EFW and AC growth velocities of >30 centiles over eight weeks compared to the remainder of the cohort, using Fisher’s exact test. This ascertained the relative risk (RR) of shoulder dystocia where an increase in centile of this magnitude was seen. Finally we compared the predictive performances of accelerated 28-36 week EFW and AC growth velocities (>30 centiles) and 36-week EFW >95th centile for shoulder dystocia. These performance parameters were calculated from all participants planned for vaginal delivery – replicating the cohort at potential risk of shoulder dystocia when seen in the antenatal clinic”

The statistical approach to data analysis is confusing. The authors intend to study the risk of shoulder dystocia according to growth velocity among fetuses with EFW < 95%ile. Therefore, they should present a univariate analysis where the baseline characteristics of the women enrolled are analyzed according to presence or absence of shoulder dystocia at birth, among these characteristics they should also describe growth velocity, both as a continuous and as a categorical variable. Then, they should build a multivariate regression model studying the relationship between the dependent variable (shoulder dystocia) and the dependent variable growth velocity (either continuous or categorical), controlling for confounding

Can the authors comment on the prevalence of shoulder dystocia, as it appears to be high for a population that does not include only diabetic mothers ?

Reviewer #2: This conceptually interesting paper by McDonald and peers examines a data set on fetal growth and the outcome of shoulder social. Specifically, this manuscript concentrates on the growth velocity of the child, which is an interesting novel marker. The paper is generally well written and the statistical approach is suitable for such an analysis. The following comments are meant to strengthen the paper:

1. The largest and potentially fatal flaw in this paper is the sample size which included 347 individuals. Though they are able to achieve statistical significance in their primary outcome this is not reassuring as it based on 6 cases of shoulder dystocia. If published the conclusion should be worded more skeptically and the need for a larger population strongly emphasized. Please note that I remain intrigued by the potential of this marker but concerned about its implementation based on the limited number of cases.

2. Though the occurrence of shoulder dystocia was objectively assessed, the clinical meaninfullness of this outcome is unclear. No comments about whether the shoulder dystocia resulted in maternal or fetal harm is discussed. Outcomes related to BPI, HIE, and maternal pelvic trauma/hemorrhage should be added.

3. Though the test performance is reported the challenge is that to avoid the shoulder dystocia significant numbers of cesareans would need to be performed (8 at 12.5% and 14 at 7.1%) to prevent one shoulder dystocia that may not result in harm as discussed in point 2. For balance the maternal risks of this should be discussed.

4. The term used of “Emergency cesarean” in table 1 & 4 I believe is better conveyed as unscheduled

5. Table 2 & 3 should be combined or woven into the text. There is not enough information to warrant a separate table.

6. The reference provided to a law suit in the UK (line 66) is fear mongering and not appropriate to include.

6. PLOS authors have the option to publish the peer review history of their article (what does this mean?). If published, this will include your full peer review and any attached files.

Reviewer #1: No

Reviewer #2: No

---

## [Author Response · Author response to Decision Letter 0]

15 Sep 2021

Please see the "Response to reviewers" letter, attached as a cover letter file, which addresses all comments made by the reviewers.

With regard to the Academic Editor's notes on formatting and style, these have all been addressed, as per the journal's guidelines, in the clean copy of the manuscript, labelled as "Manuscript"

---

## [Decision Letter · Decision Letter 1]

4 Oct 2021

Accelerated fetal growth velocity across the third trimester is associated with increased shoulder dystocia risk among fetuses who are not large-for-gestational-age: A prospective observational cohort study

PONE-D-21-00919R1

Dear Dr. MacDonald,

We’re pleased to inform you that your manuscript has been judged scientifically suitable for publication and will be formally accepted for publication once it meets all outstanding technical requirements.

Kind regards,

Kazumichi Fujioka

Academic Editor

PLOS ONE

Additional Editor Comments (optional):

Reviewers' comments:

Reviewer's Responses to Questions

**Comments to the Author**

1. If the authors have adequately addressed your comments raised in a previous round of review and you feel that this manuscript is now acceptable for publication, you may indicate that here to bypass the “Comments to the Author” section, enter your conflict of interest statement in the “Confidential to Editor” section, and submit your "Accept" recommendation.

Reviewer #2: All comments have been addressed

2. Is the manuscript technically sound, and do the data support the conclusions?

Reviewer #2: Yes

3. Has the statistical analysis been performed appropriately and rigorously? 

Reviewer #2: Yes

4. Have the authors made all data underlying the findings in their manuscript fully available?

Reviewer #2: Yes

5. Is the manuscript presented in an intelligible fashion and written in standard English?

Reviewer #2: Yes

6. Review Comments to the Author

Reviewer #2: I have no further comments. Please note that this well done.

xxxxxxxxxxxxxxxxxxxxxxxxxxxxxxxxxxxxxxxxxxxxxxxxxxxxxxxxxxxxxxxxxxxxxxxxxxxxxxxxxxxxxxxxxxxxxxxxxxxxxxxxxxxxxxxxxxxxxxxxxxxxxxxxxxxxxxxxxxxxxxxxxxxxxxxxxxxxxxxxxxxxxxxxxxxxxxxxxxxxxxxxxxxxxxxxxxxxxxxxxxxxxxxxxxxxxxxxxxxxxxxxxxxxxxxxxxxxxxxxxxxxxxxxxxxxxxxxxxxxxxxxxxxxxxxxxxxxxxxxxxxxxxxxxxxxxxxxxxxxxxxxxxxxxxxxxxxxxxxxxxxxxx

7. PLOS authors have the option to publish the peer review history of their article (what does this mean?). If published, this will include your full peer review and any attached files.

Reviewer #2: No

---

## [Editor Report · Acceptance letter]

12 Oct 2021

PONE-D-21-00919R1 

Accelerated fetal growth velocity across the third trimester is associated with increased shoulder dystocia risk among fetuses who are not large-for-gestational-age: A prospective observational cohort study 

Dear Dr. MacDonald:

I'm pleased to inform you that your manuscript has been deemed suitable for publication in PLOS ONE. Congratulations! Your manuscript is now with our production department. 

Kind regards, 

on behalf of

Dr. Kazumichi Fujioka 

Academic Editor

PLOS ONE